# ABCA1 Expression Is Upregulated in an EMT in Breast Cancer Cell Lines via MYC-Mediated De-Repression of Its Proximal Ebox Element

**DOI:** 10.3390/biomedicines10030581

**Published:** 2022-03-02

**Authors:** Sara Prijic, Jeffrey T. Chang

**Affiliations:** Department of Integrative Biology and Pharmacology, University of Texas Health Science Center at Houston, Houston, TX 77030, USA; sara.prijic@uth.tmc.edu

**Keywords:** breast cancer, epithelial–mesenchymal transition, cell motility, transcriptional regulation

## Abstract

The ATP-Binding Cassette transporter A1 (ABCA1) reverse cholesterol transport channel has been associated with a number of phenotypes in breast cancer, including reduced proliferation and increased metastatic capacity. It is induced in an epithelial–mesenchymal transition (EMT), but little is known about how this occurs, and whether it is sufficient to promote metastatic phenotypes. To address these questions, we have deciphered the transcriptional regulation of ABCA1 across EMT states and found that it is repressed by MYC via an E-box element in its P1 alternative promoter. De-repression of the promoter by MYC knockdown leads to induction of ABCA1 expression. This indicates that ABCA1 expression is regulated in an EMT, revealing another link between ABCA1 and malignant phenotypes.

## 1. Introduction

The cholesterol efflux channel ATP-Binding Cassette transporter A1 (ABCA1) has a controversial and poorly understood role in breast cancer. ABCA1 functions as a reverse cholesterol transport channel [1,2]. Autosomal recessive mutations in ABCA1 result in Tangier disease, which clinically manifests as enlarged yellow-orange tonsils due to the accumulation of cholesterol in macrophages [3,4,5]. In breast cancer cell lines, ectopic expression of ABCA1 in a mutant p53/Ras-activated background results in a decrease in proliferation [6], suggesting that it plays a tumor suppressor role. Supporting this, ABCA1 null mice develop tumors faster than wild-type mice in an MYC-activated background [7]. In contrast, ABCA1 has also been linked to malignant phenotypes. High expression has been proposed to be a marker for high grade and triple-negative breast cancer [8], and also linked to drug resistance in colon cancer [9], lung cancer [10], and melanoma [11] cells. As a prognostic factor, however, the findings are equivocal, as high expression of ABCA1 is associated with longer survival in high-grade prostate tumors [12], and both longer and shorter survival have been reported in epithelial ovarian cancer [13,14], shorter survival in neuroblastoma [15], and shorter survival in colorectal cancer [16]. Nevertheless, reports on its impact on local recurrence and metastasis more consistently point to poor outcomes. High expression of ABCA1 correlates with an increased risk of recurrence in colorectal [17] and breast cancer [18]. Its expression is also seen in an epithelial–mesenchymal transition (EMT) [18], a process that confers upon cancer cells’ increased motility, invasiveness, and ability to evade apoptosis [19], and is linked to an altered lipid composition [20]. *ABCA1* expression is high in metastatic breast cancer cell lines [20] and associated with increased metastatic capacity in mouse xenografts and breast cancer patients [18]. Taken together, ABCA1 expression appears to correlate with less tumor initiation and proliferation, but more progression and resistance, although a definitive mechanism has not yet been established.

A wealth of studies has decoded transcriptional and post-transcriptional mechanisms for the regulation of ABCA1, primarily in the context of macrophage differentiation [21]. However, despite the fact that the gene and protein expression of ABCA1 have been linked to a number of cancer phenotypes, little is known about how it is regulated in breast cancer cells. Therefore, we investigated *ABCA1* expression in breast cancer cell lines and found that it has higher expression in mesenchymal cells. Furthermore, we deciphered its transcriptional regulatory program and report that *ABCA1* expression is low in epithelial, non-metastatic breast cancer cells via MYC-mediated repression of an E-box element in its proximal promoter. This reveals the mechanism by which *ABCA1* is regulated in mesenchymal breast cancer cells and presents a potential avenue by which breast cancer cells acquire metastatic capacity.

## 2. Materials and Methods

### 2.1. Cell Lines and Cell Culture

MCF10A cells were cultured in MEBM (Lonza Bioscience, Basel, Switzerland) supplemented with MEGM BulletKit (Lonza Bioscience, Basel, Switzerland) but without GA-1000 (gentamycin-amphotericin B mix). SKBR3 cells were cultured in McCoy’s 5A (HyClone). BT474, T47D, and BT549 cells were cultured in RPMI 1640 (HyClone; Cytiva, Marlborough, MA, USA). MCF7, MDA-MB-436, and MDA-MB-231 cells were cultured in DMEM (Thermo Fisher Scientific, Waltham, MA, USA). All media were supplemented with 10% FBS (HyClone). HMLE cells were grown in a 1:1 ratio of DMEM/F-12 (Thermo Fisher Scientific, Waltham, MA, USA) and MEBM supplemented with MEGM BulletKit without GA-1000. HMLE-ER-Twist cells were induced to undergo an EMT using 20 nM 4-hydroxytamoxifen for 14 days (Sigma-Aldrich, St. Louis, MO, USA). Cells were split and medium was changed every 2–3 days.

Cells were grown on dishes with 10 cm diameter and incubated in a humidified atmosphere of 5% CO_2_ at 37 °C until they reached about 80% confluence. Medium was removed prior to washing and detaching cells with 1× PBS and 0.05% Trypsin-EDTA (Thermo Fisher Scientific, Waltham, MA, USA), respectively. An equal volume of medium with 10% FBS was added for trypsin inactivation prior to centrifuging and counting cells for subsequent experiments.

All cell lines were tested for mycoplasma using the MycoAlert Mycoplasma Detection Kit (Lonza Bioscience, Basel, Switzerland).

### 2.2. RNA Isolation, Reverse Transcription, and RT-qPCR Analysis

Total RNA was extracted with RNeasy Mini Kit (Qiagen, Hilden, Germany) after homogenizing the samples using QIAshredder (Qiagen, Hilden, Germany) and coupled with DNA digestion using the RNase-Free DNase Set (Qiagen, Hilden, Germany). Reverse transcription was performed in a T100TM Thermal Cycler (Bio-Rad, Hercules, CA, USA) using the Verso cDNA Synthesis Kit (Thermo Fisher Scientific, Waltham, MA, USA).

The obtained cDNA was added to KAPA SYBR FAST Universal 2X qPCR Master Mix (Kapa Biosystems (Roche), Basel, Switzerland) and a specific primer set. RT-qPCR was performed in a Mastercycler RealPlex2 (Eppendorf, Hamburg, Germany) with the initial denaturation at 94 °C for 2 min, followed by 40 cycles of denaturation at 94 °C for 15 s, annealing at 60 °C for 30 s, and elongation at 72 °C for 30 s. The efficiencies of every primer set were determined and taken into account when calculating gene expressions. RT-qPCR data are presented as relative gene expressions analyzed by the comparative CT method and normalized to GAPDH levels. We used the following primer sets:

GAPDH_F  GAAGGTGAAGGTCGGAGTC

GAPDH_R  GAAGATGGTGATGGGATTTC

ABCA1_F  TTAATGGGGCTGGAAAATCA

ABCA1_R  TCCTCTCAAAAGGGCAAAGA

MYC_F  CCTACCCTCTCAACGACAGC

MYC_R  CTCTGACCTTTTGCCAAGGAG

CDH1_F  AAGAAGGAGGCGGAGAAGAG

CDH1_R  TTTCCAATTTCATCGGGATT

### 2.3. Immunoblotting

To obtain proteins, media was removed, and cells were washed with chilled 1× PBS on ice prior to scrapping with Corning cell lifter (Sigma-Aldrich, St. Louis, MO, USA) in Pierce IP Lysis Buffer (Thermo Fisher Scientific, Waltham, MA, USA) with protease (Sigma-Aldrich, St. Louis, MO, USA) and phosphatase inhibitors (Sigma-Aldrich, St. Louis, MO, USA). Protein concentrations were measured using the Pierce BCA Protein Assay Kit (Thermo Fisher Scientific, Waltham, MA, USA). Of the total proteins, 10–25 μg were separated by SDS-PAGE on either 4–12% or 10% Bolt Bis-Tris Plus Gels (Thermo Fisher Scientific, Waltham, MA, USA) in Bolt MES SDS Running Buffer (Thermo Fisher Scientific, Waltham, MA, USA) at 165 V for 35–45 min at room temperature, and transferred to PVDF membranes (EMD Millipore, Burlington, MA, USA) in Bolt Transfer Buffer (Thermo Fisher Scientific, Waltham, MA, USA) at either 95 V for 2 h or 30 V overnight at 4 °C. Membranes were blocked in non-fat milk for 1 h at room temperature and probed with primary antibodies: anti-ABCA1 (Abcam #ab18180, Cambridge, United Kingdom) at 1:1000, anti-β-actin (Santa Cruz Biotechnology #sc-47778, Dallas, TX, USA) at 1:5000, anti-MYC (Santa Cruz Biotechnology #sc-40) at 1:200, anti-E-cadherin (Cell Signaling Technology #3195S, Danvers, MA, USA) at 1:1000, anti-vimentin (Cell Signaling Technology #5741S, Danvers, MA, USA) at 1:1000, and anti-GAPDH (Cell Signaling Technology #5174S, Danvers, MA, USA) at 1:5000 overnight at 4 °C. Membranes were washed 3× with PBS-T for 10 min at room temperature, and probed using either anti-mouse or anti-rabbit goat HRP-conjugated secondary antibodies (EMD Millipore, Burlington, MA, USA) for 1 h at room temperature, then washed again. Antibody-protein complexes were visualized using either non-commercial ECL-solution or SuperSignal West Femto Maximum Sensitivity Substrate (Thermo Fisher Scientific, Waltham, MA, USA).

### 2.4. ChIP-qPCR

To obtain ChIP products and genomic input DNA from HMLE/pWZL and HMLE/pWZL-Twist cells, the MAGnify Chromatin Immunoprecipitation System Kit (Thermo Fisher Scientific, Waltham, MA, USA) was used according to the manufacturer’s instructions. DNA shearing was performed for 36 min by the S220 focused-ultrasonicator (Covaris, Woburn, MA, USA) using the operating conditions recommended for the truCHIPTM low cell chromatin shearing kit with SDS shearing buffer. Then, DNA was quantified using a spectrophotometer: 30 μg of the sheared chromatin was immunoprecipitated either with 3 μg anti-MYC antibody (Abcam #ab56, Cambridge, United Kingdom) or mouse IgG while 3 μg of the chromatin was aliquoted as genomic input DNA (10% of the chromatin used for the IPs). Purified DNA from the ChIP products as well as inputs was analyzed by the qPCR performed in a Mastercycler RealPlex2 with the initial denaturation at 95 °C for 5 min, followed by 40 cycles of denaturation at 95 °C for 15 s, annealing at 60 °C for 30 s, and elongation at 72 °C for 30 s. The signals detected by the qPCR from the negative antibody control reactions, i.e., mouse IgG, were either not detected or were only present in one out of three replicates. Thus, the ChIP-qPCR data are normalized using a percent input method and presented as percentages of inputs.

The following primers were used to amplify the distal part of the proximal ABCA1 promoter (P1b), the forward primer annealing 84 bp upstream, and the reverse primer 78 bp downstream of the E-box motif:

pABCA1_F  ACACCTGCTGTACCCTCCAC

pABCA1_R  CCGAGCGCAGAGGTTACTAT

As a negative control, the following primer set amplifying region within a gene desert (GD) in chromosome 12 was used [22]:

GD_F  TGAGCATTCCAGTGATTTATTG

GD_R  AAGCAGGTAAAGGTCCATATTTC

Firefly luciferase reporter constructs

The sequences of ABCA1 promoter 1 (P1) and promoter 2 (P2) were retrieved and displayed from −500 to 200 base pairs relative to the transcription start site using the Transcriptional Regulatory Element Database, and amplified from human genomic DNA (VWR, Radnor, PA, USA) using the following primer sets:

P1_F   TGGCCTCGGCGGCCATGGCGTCCTGAGGGAGAT

P1_R  CAGTACCGGATTGCCATTGCGTCTCTTTCTCCTACC

P2_F   TGGCCTCGGCGGCCAATATCAGGTAAAATATTTTCCAAAG

P2_R  CAGTACCGGATTGCCATTACTCAGTATCTTAACCAACATG

Parts of promoter 1 were also amplified from the human genomic DNA (VWR, Radnor, PA, USA) using primer sets:

P1a_F  TGGCCTCGGCGGCCATGGCGTCCTGAGGGAGATTCAGCC 

P1a_R  CAGTACCGGATTGCCAGCCGGGGCCCGCCCCTTA

P1b_F  TGGCCTCGGCGGCCATCCACGTGCTTTCTGCTG

P1b_R  CAGTACCGGATTGCCATTGCGTCTCTTTCTCCTAC

For promoter 1, two-step PCR was performed with initial denaturation at 98 °C for 30 s followed by 35 cycles of denaturation at 98 °C for 5 s and elongation at 72 °C for 45 s. For P1a and P1b, elongation step was 30 s. For promoter 2, initial denaturation at 98 °C for 30 s was followed by 35 cycles of denaturation at 98 °C for 5 s, annealing at 60 °C for 30 s, and elongation at 72 °C for 1 min with 5% DMSO. Phusion High-Fidelity DNA Polymerase (New England Biolabs, Ipswich, MA, USA) was used for every PCR described in this manuscript.

DNA was column-purified from 0.7% agarose gel using QIAquick Gel Extraction Kit (Qiagen). Promoter sequences were cloned into the promoterless luciferase vector pGL4.10(luc2) (Promega, Madison, WI, USA) after it was linearized by digestion with HindIII-HF (New England Biolabs) using the Gibson Assembly Cloning Kit (New England Biolabs, Ipswich, MA, USA). Chemically competent One Shot TOP10 E. coli (Thermo Fisher Scientific, Waltham, MA, USA) was transformed using a heat shock and plated on agar plates with LB medium (VWR, Radnor, PA, USA) and 50 μg/mL ampicillin (VWR, Radnor, PA, USA). We verified every recombinant colony yielding a plasmid that was used for the experiments described in this manuscript by Sanger sequencing.

### 2.5. Site-Directed Mutagenesis of Firefly Luciferase Reporter Constructs

Transcription factor binding sites for E-BOX, SP1, and LXR motif of the region from −150 to 351 in the distal ABCA1 promoter 1 sequence (P1b) were identified using the JASPAR database [23] and mutations were introduced as follows:

E-BOX  CACGTG > CACGAG

SP1   GGGGCGGGG > GGCTCGCTG

LXR   TGACCGATAGTAACCT > TGAAGGATAGTAAAGT

Site-directed mutagenesis on transcription factor binding sites for E-BOX, SP1, and LXR in the distal ABCA1 promoter 1 sequence (P1b) was performed by two-step PCR with initial denaturation at 98 °C for 30 s followed by 35 cycles of denaturation at 98 °C for 30 s and elongation at 72 °C for 45 s using the following primer sets:

E-BOX-mut_F  GCCTCCACGAGCTTTCTGCTGAGTGACTGA

E-BOX-mut_R  AAAGCTCGTGGAGGCCGCCGAGGCCAGATC

SP1-mut_F  AAGGGCTCGCTGAGGAGGGAGAGCACAGGC

SP1-mut_R  GCCTGTGCTCTCCCTCCTCAGCGAGCCCTT

LXR-mut_F  TGAAGGATAGTAAAGTCTGCGCTCGGTGCA

LXR-mut_R  ACTTTACTATCCTTCAAAGCCTGTGCTCTC

### 2.6. Transient Transfection and Luminescence Measurement

To measure exogenous ABCA1 promoter activity via luminescence, 8 × 10^4^ HMLE, HMLE/pWZL-Twist, MCF7, or MDA-MB-231 cells per well were seeded in 0.5 mL medium onto 24-well plates. After 24 h, cells were co-transfected with 0.5 μg pGL4.10[luc2] experimental firefly luciferase reporters and 0.5 μg pRL-SV40 control renilla luciferase vector (Promega, Madison, WI, USA) using Lipofectamine 3000 Reagent (Thermo Fisher Scientific, Radnor, PA, USA). After 48 h, luminescence was measured with Base Infinite M200 Pro Microplate Reader (Tecan Group Ltd., Männedorf, Switzerland) using a Dual-Luciferase Reporter Assay System (Promega, Madison, WI, USA). Expression of the experimental firefly luciferase gene was normalized to the expression of control renilla luciferase gene.

### 2.7. Retroviral Construct and ABCA1 Overexpression

ABCA1 was amplified from human ABCA1 (NP_005493) VersaClone cDNA plasmid (R&D Systems, Minneapolis, MN, USA) by performing a two-step PCR with initial denaturation at 98 °C for 2 min followed by 35 cycles of denaturation at 98 °C for 30 s and elongation at 68 °C for 7 min using the following primer set:

_F  GATGTGGTGGTACGTAGGATGGCTTGTTGGCCTCAG

_R  TGGAAAATAACCGGAATTGGTCATACATAGCTTTCTTTCACTTTC

PCR product was column-purified from 0.7% agarose gel using QIAquick Gel Extraction Kit, and cloned into expression retroviral vector pWZL Hygro, a gift from Scott Lowe (Addgene plasmid #18750), after it was linearized by digestion with EcoRI-HF and SalI (both New England Biolabs, Ipswich, MA, USA) using the Gibson Assembly Cloning Kit. ElectroMAX DH5α-E cells (Thermo Fisher Scientific, Radnor, PA, USA) were transformed using Eppendorf Eporator (Eppendorf, Hamburg, Germany). Total length of the clone was verified by sequencing.

To create amphotropic retroviruses, packaging Platinum-A (Plat-A) cells were grown in 8 mL of DMEM on dishes with 10 cm diameter and transfected with 4 μg of either pWZL or pWZL-ABCA1 plasmids using Lipofectamine 3000 Reagent. After 48 h, viral supernatant was harvested and filtered through 0.45 μm pore size filter (Thermo Fisher Scientific, Radnor, PA, USA) prior to adding 8 μg/mL polybrene (Sigma-Aldrich, St. Louis, MO, USA) to the supernatant to improve infection efficiency. Cells of the target cell line MCF7 were incubated with viral supernatant overnight before it was replaced with the fresh medium. After 24 h, we started the selection of transduced cells by adding 200 μg/mL hygromycin B (Thermo Fisher Scientific, Radnor, PA, USA) to the medium for 4 weeks.

### 2.8. RNA Interference

To knockdown MYC in HMLE and MCF7 cells, we performed reverse transfection on 2 × 10^5^ cells per well that were plated on a 6-well plate in 2 mL medium. Cells were transfected with 50 nM siRNA and 0.8% (HMLE) or 2% (MCF7) Lipofectamine RNAiMAX Reagent (Thermo Fisher Scientific, Radnor, PA, USA). To knockdown MYC and ABCA1 simultaneously in HMLE and MCF7 cells, cells were transfected with 25 nM siRNA against MYC and 25 nM siRNA against ABCA1. After 72 h, total proteins and RNA were collected for immunoblotting and qPCR, respectively. We used the following oligos (Sigma-Aldrich, St. Louis, MO, USA):

siCON  MISSION siRNA Universal Negative Control #1

siMYC #2   SASI_Hs01_00222677

siMYC #3   SASI_Hs01_00222678

siMYC #5   SASI_Hs01_00333889

siABCA1 #1  SASI_Hs01_00129036

siABCA1 #2  SASI_Hs01_00129037

### 2.9. Migration Assays

To measure cell migration after ABCA1 overexpression, knockdown of MYC or MYC and ABCA1 simultaneously, and after knockout of ABCA1, cells were put in starvation medium overnight (DMEM/F-12 for HMLE, DMEM 0.1% FBS for MCF7, and MDA-MB-231). Then, 1 × 10^5^ cells in 0.5 mL starvation medium were transferred into the upper chamber of a Thinsert Cell Culture Inserts with 8 μm pores (Greiner Bio-One, Monroe, NC, USA) in a 12-well plate in triplicates while 1.5 mL of the complete medium was added to the bottom chamber. After 16 h, cells were removed from the top surface of the upper chamber by scrubbing it twice with a cotton swab. We fixed migrated cells on the bottom surface of the upper chamber with 100% methanol for 10 min at room temperature. Nuclei of migrated cells were stained with either 1 μg/mL DAPI for 2 min or 0.05% crystal violet for 15 min at 37 °C. After DAPI staining, photos of migrated cells in 9–12 visual fields were taken while observing them using the digital inverted microscope EVOS fl with fluorescence light application at 10× objective. For quantification purposes, photos were converted into binary images and analyzed for particles using Image 1J 1.50i [24]. The average particle number for every experimental group was calculated from 9 to 12 visual fields and normalized to the average particle number of the control group. Representative images for every experimental group are shown. After crystal violet staining, cells in 9 visual fields were counted while observing them using an inverted Nikon Eclipse TE2000-U microscope. The average number of migrated cells for every experimental group was calculated from 9 visual fields and normalized to the average number of the cells that migrated in the control group. Representative images for every experimental group are shown.

### 2.10. Cellular Cholesterol Assays

To measure cellular cholesterol in HMLE cells after knockdown of MYC or MYC and ABCA1 simultaneously, we performed reverse transfection on a 6-well plate as described above. After 48 h, the complete medium was removed, and cells were put in starvation DMEM/F-12 medium overnight. To measure cellular cholesterol after overexpression of ABCA1, 2 × 10^5^ cells were seeded on a 6-well plate. After 24 h, cells were trypsinized, centrifuged, resuspended in PBS, and counted. We measured cholesterol content as described previously [18] and normalized it to the amount of protein that were measured using the Pierce BCA Protein Assay Kit (Pierce Biotechnology, Waltham, MA, USA).

For cholesterol treatment, HMLE cells were supplemented with 500 nM water soluble cholesterol (Sigma-Aldrich #C4951, St. Louis, MO, USA) every day for 3 consecutive days.

### 2.11. Membrane Fluidity Assays

To measure membrane fluidity after knockdown of MYC or MYC and ABCA1 simultaneously, we performed reverse transfection on a 96-well plate as described above. After 48 h, the full media was changed to starvation media overnight. Membrane fluidity was measured using a Marker Gene Technologies kit (Marker Gene Technologies, Eugene, OR, USA) as described previously [18]. 

### 2.12. CRISPR/Cas9

The CRISPR design web tool at crispr.mit.edu was used to obtain sgRNA sequences against ABCA1 promoter 1 and exon 3. We chose sgRNAs with the highest score and the lowest number of off-target sites, especially in a gene coding region (PAM sequences not included):

sgRNA1  GGCTGAACGTCGCCCGTTTA targeting ABCA1 promoter

sgRNA2  GCCGAACAGATCAGGATC targeting ABCA1 exon 3

Complementary oligos with belonging overhangs were designed and sgRNAs were cloned into the plasmids using Golden Gate cloning protocol [25]. The sgRNA1 was cloned into lenti sgRNA(MS2)_zeo backbone (Addgene plasmid #61427, Watertown, MA, USA) [26] after digestion with BsmbI (New England Biolabs, Ipswich, MA, USA). The sgRNA2 was cloned into pGL3-U6-sgRNA-PGK-puromycin (Addgene plasmid #51133, Watertown, MA, USA) [27] after digestion with Eco31l (Thermo Fisher Scientific, Radnor, PA, USA). Positive clones were verified for sgRNAs insertion by sequencing.

We made high specificity pSpCas9(1.1)-2A-GFP by inserting 2A-GFP into eSpCas9(1.1) (Addgene plasmid #71814, Watertown, MA, USA) [28] after it was linearized with EcoRI (New England Biolabs, Ipswich, MA, USA). To obtain 2A-GFP PCR product from the pSpCas9n(BB)-2A-GFP (PX461) (Addgene plasmid #48140, Watertown, MA, USA) [25], a two-step PCR with initial denaturation at 98 °C for 30 s followed by 35 cycles of denaturation at 98 °C for 5 s and elongation at 68 °C for 1.5 min was performed using the following primers:

2A-GFP_F  GAAAAAGTCAGAATTCGGCAGTGGAGAGGGCAGAGG

2A-GFP_R  CGAGCTCTAGGAATTCTTAGAATTCCTTGTACAGCTCGTCCA

DNA was column-purified from 0.7% agarose gel using the QIAquick Gel Extraction Kit and cloned into eSpCas9(1.1) using the Gibson Assembly Cloning Kit. ElectroMAX DH5α-E cells were transformed using an Eppendorf Eporator, and the insert in the recombinant clone was verified by sequencing.

Then, 2.5 × 10^6^ MDA-MB-231 cells were transfected with either 2.5 μg sgRNA(MS2)_zeo backbone-sgRNA1 (g1 clone) or pGL3-U6-sgRNA-PGK-puromycin-sgRNA2 (g2 clone) using Lipofectamine 3000 on a 6-well plate according to the manufacturer’s recommendations. Transfected cells were selected with either 400 μg/mL zeocin for 2 weeks (g1) or 1 μg/mL puromycin for 3 days (g2). After selection, 2.5 × 10^6^ MDA-MB-231/g1 and MDA-MB-231/g2 cells were transfected with 2.5 μg pSpCas9(1.1)-2A-GFP using Lipofectamine 3000 on a 6-well plate according to the manufacturer’s recommendations. After 48 h, GFP+ cells were selected using BD FACSAria II (BD Biosciences, Franklin Lakes, NJ, USA). GFP+ cells were expanded following an in-house single-cell dilution protocol. After single cell clonal expansion, proteins were collected, and cells were tested for ABCA1 expression as described above. GAPDH was used as a loading control.

### 2.13. Statistical Analysis

All quantitative data are presented as means and standard deviations. Student’s t-test was used to assess the differences between the experimental groups followed by the Holm–Sidak method to correct for multiple comparisons. The alpha level was set to 0.05. When *p* values were 0.05 or less, differences were considered statistically significant. All statistical analyses were performed using the BETSY system [29] or GraphPad Prism 8 for Mac OS X (GraphPad Software, San Diego, CA, USA). 

## 3. Results

### 3.1. ABCA1 Is Transcriptionally Regulated in Mesenchymal Breast Cancer Cell Lines

We measured the gene expression of ABCA1 in a panel of breast cell lines, including a non-cancer breast epithelial outgrowth MCF10A [30], luminal-type breast cancer cells that exhibit an epithelial phenotype (SKBR3, BT474, MCF7, and T47D), and basal b-type breast cancer cells that are mesenchymal (MDA-MB-436, MDA-MB-231, and BT-549) [31] (Figure 1a). The luminal cells were previously shown to be non-metastatic, while the basal cells could metastasize efficiently in mice [32]. In this panel of cells, the expression of ABCA1 was correlated with the cell state, with markedly higher expression in the mesenchymal basal B cells.

Seeing a correlation of ABCA1 expression with the mesenchymal phenotype, we next asked whether this was sufficient to induce changes in cell motility, which is acquired during an EMT. We have previously shown that stable knockdown of *ABCA1* decreases the motility of the mesenchymal MDA-MB-231 cells [18]. We confirmed here that CRISPR knockout can also increase cholesterol, decrease plasma membrane fluidity, and decrease migration (Appendix A). However, it is not yet known whether ectopic expression of ABCA1 can increase motility. To address this, we stably expressed ABCA1 in MCF7 cells, a cell line with low endogenous expression (Figure 1b). These transgenic cell lines showed an increase in migration (Figure 1c,d), an increase in membrane fluidity (Figure 1e), and loss of cellular cholesterol (Figure 1f). Each of these experiments were repeated a minimum of three times. This reveals that expression of ABCA1 is associated with a decrease in cellular cholesterol, consistent with its efflux activity, as well as a concomitant increase in cell migration.

### 3.2. E-Box and SP1 Promoter Elements Regulate ABCA1 in an EMT

Seeing a marked difference in the expression of ABCA1 in the epithelial and mesenchymal cell lines, we next dissected the transcriptional regulation of *ABCA1* by cloning its two alternative promoters, P1 and P2, that were previously reported to contain regulatory elements [21] into luciferase reporter plasmids (Figure 2a). After transfecting these plasmids into isogenic HMLE and HMLE-Twist cells [33,34], representing the epithelial or mesenchymal phenotypes, respectively, we saw that promoter P1 was associated with increased luciferase expression in the Twist-expressing mesenchymal cells (Figure 2b), while promoter P2 exhibited little or no activity in either of these conditions. To further narrow down the regulatory region, we created a vector covering a more limited region of the proximal promoter, which we called P1b. By luminescence assay, we saw that the P1b region alone could induce transcriptional activity nearly as much as the full promoter (Figure 2c). Therefore, we concluded that transcriptional regulation of *ABCA1* expression in the epithelial/mesenchymal state occurs in the P1b proximal promoter region.

Within the P1b promoter region, there are several documented regulatory elements including E-box, SP1, and LXR elements [21]. To isolate regulation to a specific element, we mutated each one using site-directed mutagenesis (Figure 2d). We then transfected these constructs into four cell lines—MCF7 and HMLE, representing the epithelial state, and MDA-MB-231 and HMLE-Twist as mesenchymal—and measured luminescence (Figure 2e). These experiments were repeated a minimum of three times. This showed that mutations in the LXR element reduced transcription in all contexts, confirming its established role as a strong activator of *ABCA1* expression [35,36]. However, this regulation was seen across both cancer cell phenotypes, and thus cannot account for the differences in the expression of *ABCA1* in an EMT.

In contrast to LXR, mutations in the E-box and SP1 elements do exhibit state-specific differences in regulation, where E-box mutation leads to de-repression of *ABCA1* in the epithelial state, and SP1 leads to repression in the mesenchymal state (and potentially also in the epithelial, although it did not achieve statistical significance). We interpret this to mean that an E-box binding transcription factor represses *ABCA1* expression in the epithelial state, while an SP1 binding factor induces *ABCA1* expression in the mesenchymal state. However, because the E-box mutation is associated with a larger overall change in expression, we elected to follow up on this element.

### 3.3. MYC Binds to the ABCA1 Promoter and Represses Its Expression

E-box elements can be bound by a range of basic helix–loop–helix transcription factors [37], and the motif in the *ABCA1* promoter was previously reported to be repressed by FOSL2 in macrophages [38]. To predict the transcription factor that is most likely to bind this element in epithelial cells, we took a bioinformatics approach and analyzed the gene expression profiles of a panel of 74 breast cancer cells in either an epithelial or mesenchymal state from six data sets that we previously described [18]. We then correlated the expression of candidate E-box binding transcription factors (USF1/2, TFE3, TFEB, MYC, SREBP1/2, MITF, NRF1, MLXIPL, ZEB1/2, SNAI1, SNAI2, HIF1A, FOSL2, BMAL1, and MYOD1) with the epithelial or mesenchymal states (Figure 3a). As controls, we included the expression of EMT marker genes *CDH1*, *CDH2*, and *VIM* [39]. We confirmed that the epithelial-expressed cadherin *CDH1* expression is correlated with the epithelial state (*p* = 7.6 × 10^−9^), and mesenchymal cadherin *CDH2* (*p* = 1.3 × 10^−10^) and mesenchymal marker *VIM* are correlated with the mesenchymal state (*p* = 1.2 × 10^−7^). Among the transcription factors, two were most highly correlated with the cell states: *ZEB1*, with high expression in mesenchymal cells (*p* = 7.6 × 10^−9^), and *MYC*, expressed highly in epithelial cells (*p* = 4.3 × 10^−8^). Because the prior experiments suggested that the E-box was repressed in the epithelial state, we predicted from the expression patterns that MYC may repress *ABCA1* in epithelial breast cancer cells.

To confirm the prediction that *MYC* expression is higher in the epithelial state than mesenchymal, we started by testing the expression of *ABCA1* in HMLE cells, an epithelial breast cell line that is amenable to an EMT. We induced an EMT by expression of the EMT inducer TWIST1 and confirmed that it was higher in the EMT-induced mesenchymal state than the baseline epithelial one (Figure 3b). Then, we performed a time course experiment where we induced HMLE cells expressing an ER-inducible TWIST1 construct to undergo an EMT by treatment with 4-hydroxytamoxifen and monitored the expression of *MYC* throughout this transition (Figure 3c). At the end of 14 days, we observed decreased *CDH1* and increased *VIM* expression, confirming an EMT. We also saw increased expression of *ABCA1*, as well as loss of *MYC*, confirming the bioinformatic prediction that *MYC* is expressed specifically in the epithelial state.

To determine whether MYC regulates the expression of *ABCA1*, we knocked down the expression of *MYC* in HMLE cells with three siRNA sequences and saw a loss of *MYC* concomitant with increased gene and protein expression of *ABCA1* (Figure 3d). Then, to ascertain whether MYC regulates *ABCA1* by binding directly onto the *ABCA1* promoter, or exerts its effect through other, potentially indirect, mechanisms, we performed a chromatin immunoprecipitation experiment and found that the binding of MYC to the *ABCA1* promoter (Figure 3e) is decreased by 2.4-fold in mesenchymal cells (HMLE-Twist) relative to epithelial (HMLE). Biological replicates were performed as described in the figure legend. These experiments demonstrate that MYC binds directly to the proximal promoter of *ABCA1*, that the binding affinity is decreased in mesenchymal cells, and that loss of MYC leads to induction of *ABCA1* expression, supporting a model where MYC can repress *ABCA1* expression in the epithelial state.

## 4. Discussion

The role of the cholesterol efflux channel ABCA1 in cancer is controversial, having been associated with a range of sometimes conflicting phenotypes across diverse conditions and cancer types. This is potentially due to pleiotropic impacts on a range of phenotypes, such as cell proliferation or migration, that have differing impacts on tumor initiation, progression, or metastasis. To understand its role across this diversity of conditions, its regulation and function must be carefully dissected.

In this study, we demonstrated that *ABCA1* expression is highly regulated across cancer cell states. In some experiments, we leveraged an isolated model of EMT that is induced by TWIST1 and associated with decreased *MYC* and increased *ABCA1* expression. However, the regulation network amongst these proteins is likely to be more complex. *TWIST1* was originally identified to be induced by MYC in *Xenopus* development [40], which, together with our observations, suggests that it may serve as a negative feedback mechanism to prevent the deleterious consequences of excessive MYC expression. In fact, loss of *TWIST1* function exacerbates MYC-induced apoptosis and has been linked to developmental defects associated with apoptosis [41].

The complexity of the network highlights a limitation of this study in that only a selected number of cell lines were tested, and that an in vivo validation of the regulation in the pathway is lacking. It is possible that additional observed co-factors may be involved in this regulation. Nevertheless, the context-dependent expression of ABCA1 may provide a partial explanation for the conflicting reports regarding its impact on outcomes, as the importance of these phenotypes to the overall malignancy of the disease may vary across contexts. In breast cancer, cells can be plastic and convert between epithelial and mesenchymal states, where they exhibit differing levels of ABCA1 expression. The EMT has previously been associated with additional malignant phenotypes, including stem cell phenotypes, chemoresistance, and immune evasion. However, it is not known whether these phenotypes are dependent upon ABCA1 expression, which should be investigated in follow-up experiments. More work is needed to dissect the multifaceted role of ABCA1 in breast cancer, as well as to understand its contribution to the lipid metabolic reprogramming that seen across a range of cancer states. A better understanding of this phenomenon may lead to novel therapeutic avenues to control disease. 

## Figures and Tables

**Figure 1 biomedicines-10-00581-f001:**
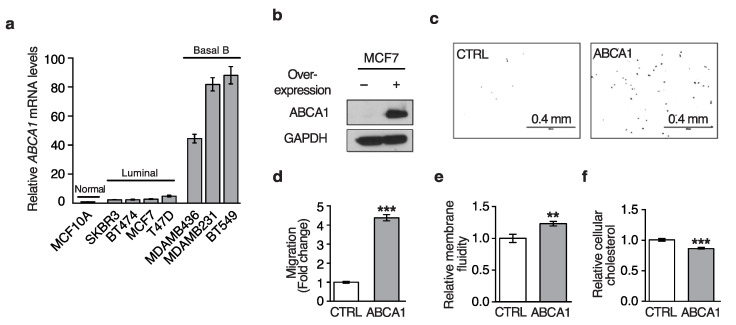
ABCA1 expression is higher in mesenchymal breast cancer cells and promotes migration in MCF7 cells. (**a**) Relative gene expression levels of *ABCA1* are shown on the *y*-axis across breast cell lines. *Normal* indicates a non-cancer cell line, and *Luminal* and *Basal B* indicates the subtype of the cancer cell lines. Error bars indicate one standard deviation. (**b**) This immunoblot shows the expression of ABCA1 and GAPDH in MCF7 cells with or without overexpression of ABCA1. (**c**) A representative image (from 15 fields) shows the migrated MCF7 cells in from a transwell migration assay for the baseline condition (**left**) as well as those with ABCA1 expression (**right**). (**d**) The relative number of migrated cells are shown on the *y*-axis for the two conditions. Error bars indicate one standard deviation. Statistical significance, relative to the CTRL condition, is indicated by ** *p* < 0.01, *** *p* < 0.001. (**e**) The relative membrane fluidity is shown on the *y*-axis (*n* = 3 technical replicates). Error bars indicate one standard deviation. (**f**) The relative cellular cholesterol content is shown on the *y*-axis (*n* = 3 technical replicates). Error bars indicate one standard deviation. At least three biological replicates were performed for each experiment, and representative data are shown.

**Figure 2 biomedicines-10-00581-f002:**
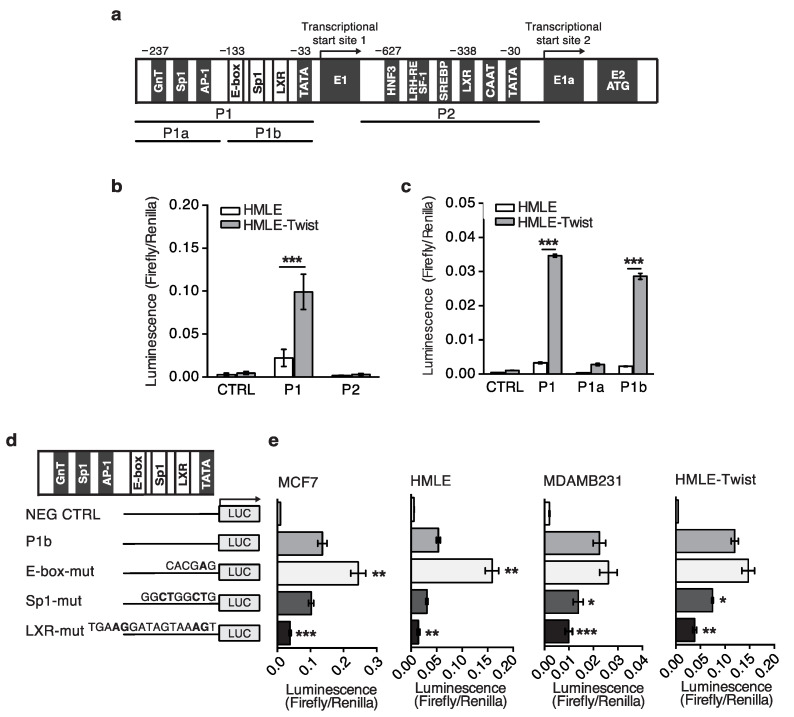
*ABCA1* expression is differentially regulated in mesenchymal cell lines through the E-box motif in its proximal promoter. (**a**) This schematic outlines the structure of *ABCA1* alternative promoters. (**b**) Luminescence induced by alternative proximal promoters P1 or P2 are shown on the *y*-axis (*n* = 4 technical replicates). Error bars indicate one standard deviation. Statistical significance is indicated as in Figure 1. (**c**) Luminescence induced by promoter P1, or promoter fragments P1a and P1b, are shown on the *y*-axis (*n* = 4 technical replicates). Error bars indicate one standard deviation. (**d**) This shows the mutations introduced to the P1b promoter fragment to disrupt binding capacity of the E-box, SP1, and LXR motifs. Mutations are labeled in bold. (**e**) Luminescence is shown on the *x*-axis for each of the mutant promoters in the MCF7, HMLE, MDA-MB-231, and HMLE-Twist cell lines (*n* = 4 technical replicates, for each). Error bars indicate one standard deviation. At least three biological replicates were performed for each experiment, and representative data are shown. * *p* < 0.05, ** *p* < 0.01, *** *p* < 0.001.

**Figure 3 biomedicines-10-00581-f003:**
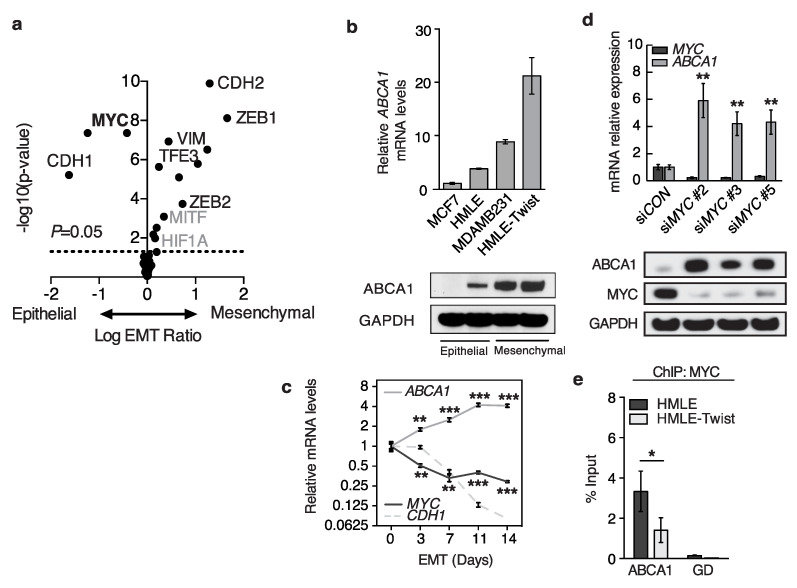
MYC binds to the *ABCA1* promoter and represses its expression in epithelial cells. (**a**) This volcano plot shows the relative log_2_ expression of E-box binding transcription factors on the *x*-axis. Each transcription factor is shown as a dot, and those expressed higher in mesenchymal cells are shown on the right, while those expressed higher in epithelial cells are on the left. The *y*-axis shows the −log_10_ of the *p*-value. A dotted line indicates *p* = 0.05. (**b**) The gene (**top**) and protein (**bottom**) expressions of ABCA1 are shown for four cell lines (*n* = 3 technical replicates). Error bars indicate one standard deviation. Statistical significance is indicated as in Figure 1. (**c**) The relative gene expressions of *ABCA1*, *MYC*, or *CDH1* are shown on the *y*-axis across time (*x*-axis) in log scale (*n* = 3 technical replicates). Error bars indicate one standard deviation. (**d**) (**top panel**) The relative gene expressions of *ABCA1* or *MYC* in HMLE cells are shown on the *y*-axis after knockdown with three independent siRNAs targeting *MYC* (*n* = 3 technical replicates). Error bars indicate one standard deviation. (**bottom**) These immunoblots show the protein expression in the same conditions. (**e**) The binding affinity of MYC to the *ABCA1* promoter or a gene desert (*GD*) region is quantified relative to input (*y*-axis) (*n* = 3 technical replicates). Error bars indicate one standard deviation. At least three biological replicates were performed for experiments shown in panels b, d, and e, and representative data are shown. The data in panel (**a**) were aggregated from six data sets. For panel (**c**), the time series has been measured over five times, but the time points profiled in the middle samples varied. * *p* < 0.05, ** *p* < 0.01, *** *p* < 0.001.

## Data Availability

Not applicable.

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
