# Peer review of "ABCA1 Expression Is Upregulated in an EMT in Breast Cancer Cell Lines via MYC-Mediated De-Repression of Its Proximal Ebox Element"

_biomedicines, 2022, doi:10.3390/biomedicines10030581_

Round 1

Reviewer 1 Report

The manuscript « The manuscript "ABCA1 expression is upregulated in an EMT 2 via MYC-mediated de-repression of its proximal Ebox element” by Sara Prijic and Jeffrey T. Chang presents data demonstrating that the cholesterol transporter ABCA1 is upregulated in EMT in breast cancer cells via MYC de-repression of the proximal Ebox element in the promoter.

The paper is well-written and easy to follow. The experiments are well conducted and the message is straightforward.

The main criticism with this manuscript is that it looks more like a Master student report or a brief communication than a real full scientific article. The message is clear and straightforward but the significance of this work at this stage is unclear. This dataset would merit in my view to be published in a wider article.

Minor comment: title should mention breast cancer

Author Response

The main criticism with this manuscript is that it looks more like a Master student report or a brief communication than a real full scientific article. The message is clear and straightforward but the significance of this work at this stage is unclear. This dataset would merit in my view to be published in a wider article.

Thank you for your comments, and your appreciation of the clarity of the story.  We agree that the scope of the article here is limited.  However, we feel that a better understanding of the mechanism by which ABCA1 is regulated in cancer cells, which is somewhat different than what has been previously reported in myeloid cells, is an important contribution that can help to shed light on the regulatory networks that help to induce expression of the gene, which can reveal deeper insight into the role that it plays in the cancer phenotype.

Minor comment: title should mention breast cancer

Thank you for the suggestion.  We have revised the title to:

ABCA1 expression is upregulated in an EMT in breast cancer cell lines via MYC-mediated de-repression of its proximal Ebox element

Reviewer 2 Report

Dr. Chang’s group previously reported that the compounds potentially inhibit EMT identified by in silico screening suppress the metastatic capacity of breast cancer cells through upregulation of ABCA1. In this study, authors examined a mechanism for ABCA1 upregulation in EMT. They found that expression levels of ABCA1 are higher in metastatic, mesenchymal breast cancer cell lines compared to non-metastatic, its luminal type. Reporter assays showed that E-box in the promoter region of ABCA1 is associated with de-repression of ABCA1 and MYC binding to the E-box in the ABCA1 promoter is significantly reduced in EMT induced breast cancer cells. The level of MYC mRNA is higher in an epithelial state of breast cancer cells than that in a mesenchymal state and is decreased upon induction of EMT. ABCA1 level was inversely correlated with MYC level during EMT. Based on these results, authors concluded that MYC represses ABCA1 expression in epithelial state and the reduction of MYC in EMT results in de-repression of ABCA1. This study is well conducted and results are clearly shown and significant to the field. I recommend this article to be published in BioMedicines. I have a few suggestions to strengthen their conclusions.

(1) In figure.3, it is clearly shown that the knockdown of MYC by siRNA results in increased levels of ABCA1 in the EMT induced cells and that induction of EMT reduces MYC levels. Have you examined the effect of overexpression (or re-expression) of MYC in EMT induced cells? Does MYC expression in EMT cells reduce ABCA1?

(2) In relation with (1), does re-introduction of MYC to EMT cells reduce ABCA1 or metastasis related features such as cholesterol levels, the cell motility etc.?

Author Response

(1) In figure.3, it is clearly shown that the knockdown of MYC by siRNA results in increased levels of ABCA1 in the EMT induced cells and that induction of EMT reduces MYC levels. Have you examined the effect of overexpression (or re-expression) of MYC in EMT induced cells? Does MYC expression in EMT cells reduce ABCA1?

Yes, this is an important experiment that would help to demonstrate that not only is MYC necessary to induce ABCA1 during the epithelial state, but also that it is sufficient to repress it during the mesenchymal state.  This is also something that we would like to explore.

However, in our previous experiments, ectopic expression of MYC induces apoptosis, possibly due to the serum-free media used for the HMLE cells.  To perform an experiment that is not confounded, we would need to limit apoptosis by either calibrating the expression of MYC or modifying the growth conditions to repress MYC-mediated apoptosis.  While these experiments would be a powerful demonstration of the concept of MYC-regulation of ABCA1, because they are not straightforward to perform technically, and because it is not clear that MYC repression of ABCA1 is a physiological event that is critical for breast cancer cell lines or tumors, we have not yet performed these experiments.

(2) In relation with (1), does re-introduction of MYC to EMT cells reduce ABCA1 or metastasis related features such as cholesterol levels, the cell motility etc.?

Yes, we desperately wish we had a system set up that could test this (as described above)!

This manuscript is a resubmission of an earlier submission. The following is a list of the peer review reports and author responses from that submission.

Round 1

Reviewer 1 Report

Major points:

  1. Missing Figure 4 for section 3.4.
  2. Please discuss the potential regulatory roles (directly or indirectly) of Twist in the proposed Myc-ABCA1-cholesterol axis.

Minor points:

  1. Typo, line 69: Cell lines and “culture culture”
  2. Typo, line 128: “ChIP-RT-qPCR.” There are no RT reactions required in the described experiments in this section.
  3. Please correct the format: “2 ✕ 105 cells.” Similarly, in line 256, line 276, line 289, line 329, and line 333.
  4. Typo, line 261: “Nuclea.” Correct form: “Nuclei.”
  5. Please specify the number of both biological repeats and experimental repeats for all the applicable experiments (qPCR, migration assay, membrane fluidity, cellular cholesterol content, luminescence assays, and Western blotting) and representative images in the Figure Legends.
  6. Please consistently use either “MDA-MB-231” or “”MDAMB231” in the manuscript.
  7. Line 366. It is too strong to claim loss of cellular cholesterol with the reported results. Please tone it down.

Reviewer 2 Report

In this manuscript, the ATP-Binding Cassette transporter A1 (ABCA1) enhanced the migration of breast tumor cells. The transcript factor, MYC, regulated the ABCA1 promoter. The data are too preliminary to demonstrate the results.

  1. The animal experiments should be shown in the manuscript.
  2. The migration assay in the knockdown of ABCA1 in BT549 cells showed be shown in the manuscript.
  3. The migration assay in different MYC expression cells should be demonstrated in the manuscripts.
  4. The cell proliferation assay also should be demonstrated in the manuscript. The MYC regulated the cell cycle.

Round 2

Reviewer 1 Report

Major points:

  1. Main text, Legend, and Results for the Figure 4 can be improved. The overall interpretation for the results is insufficient. The authors should explain why they use only the specific cell line(s) for certain experiments and not the others. How does siABCA1 or adding cholesterol alone affect the results/interpretations? The experiments for Figure 4 should include more cell lines, at least one luminal and one mesenchymal cell lines. Additionally, the “proliferation” assay was done by measuring cell “viability.” Although this is commonly seen in the literature, but it will be nice to further perform cell counting at different days to confirm these results. I would prefer the authors label the graphs “Cell viability” instead of “Proliferation.” Since the “viability” was affected by siMYC so much, the authors should also show bright-field or crystal violet-stained images of the cells in different treatment groups, so we can clearly see the cell morphology and appropriately interpret the results.
  2. Please clearly explain what the "n" number stands for in each experiment. Biological or experimental repeats?
  3. For each assay, please clearly say what cell line was used and why use such a cell line.

Minor points:

  1. Lacking sample number for Fig 3b.
  2. Error bar represents SD or SE? Please clarify this in the text.
  3. Line 453, typo: Figure 4b is meant to be Figure 4c.
  4. Line 454, typo: Figure 4c is meant to be Figure 4d.
  5. Line 459, typo: Figure 4d is meant to be Figure 4e.
  6. Line 459-463, please add “(Figure 4, f and g)” or another appropriate format.

Reviewer 2 Report

The revised manuscript did not fully response our questions.

Author Response

We have checked the review as well as our point-by-point response.  We are not sure how we failed to respond to the criticisms fully.  We are willing to try to resolve the concerns.

Round 3

Reviewer 1 Report

While the authors present an interesting MYC-ABCA1 regulatory mechanism in breast (cancer) cell migration. The experimental results shown in the manuscript are not sufficient to support their claims. I list major issues here:

  1. Lacking biological repeats. The authors only presented results from technical repeats. At least three biological repeats (in addition to multiple technical repeats) are required for all experiments unless the authors observed a more than 4-fold difference between experimental conditions.  
  2. If the authors do not include a mesenchymal cell line in Figure 4, then they should include a Western blotting results for MYC and ABCA1 for all the cell lines included in the current Fig. 1a.

Reviewer 2 Report

The content of manuscript is not ready in Page 11.